# Harnessing Inorganic Nanoparticles to Direct Macrophage Polarization for Skeletal Muscle Regeneration

**DOI:** 10.3390/nano10101963

**Published:** 2020-10-02

**Authors:** Francesca Corsi, Felicia Carotenuto, Paolo Di Nardo, Laura Teodori

**Affiliations:** 1Department of Fusion and Technologies for Nuclear Safety and Security, Diagnostic and Metrology (FSN-TECFIS-DIM), ENEA, 00044 Frascati, Italy; francesca.corsi@uniroma2.it (F.C.); carotenuto@med.uniroma2.it (F.C.); 2Department of Clinical Science and Translational Medicine, University of Rome “Tor Vergata”, 00133 Rome, Italy; dinardo@uniroma2.it; 3Interdepartmental Center of Regenerative Medicine (CIMER), University of Rome “Tor Vergata”, 00133 Rome, Italy; 4L.L. Levshin Institute of Cluster Oncology, I. M. Sechenov First Medical University, 119991 Moscow, Russia

**Keywords:** nanotechnology, tissue engineering, immunomodulation, macrophage plasticity, skeletal muscle regeneration

## Abstract

Modulation of macrophage plasticity is emerging as a successful strategy in tissue engineering (TE) to control the immune response elicited by the implanted material. Indeed, one major determinant of success in regenerating tissues and organs is to achieve the correct balance between immune pro-inflammatory and pro-resolution players. In recent years, nanoparticle-mediated macrophage polarization towards the pro- or anti-inflammatory subtypes is gaining increasing interest in the biomedical field. In TE, despite significant progress in the use of nanomaterials, the full potential of nanoparticles as effective immunomodulators has not yet been completely realized. This work discusses the contribution that nanotechnology gives to TE applications, helping native or synthetic scaffolds to direct macrophage polarization; here, three bioactive metallic and ceramic nanoparticles (gold, titanium oxide, and cerium oxide nanoparticles) are proposed as potential valuable tools to trigger skeletal muscle regeneration.

## 1. Introduction

Tissue engineering (TE) is a multidisciplinary field including bio-medicine, material science, and engineering, aimed at manufacturing functional biological tissues to be implanted in organs damaged by otherwise incurable diseases or severe casualties [1,2]. Engineered tissues are also fundamental to set up in vitro models of human physiological systems, replacing animal models for drug development, toxicology studies, etc. [3].

A successful TE treatment depends on the immune response of the recipient tissue. In fact, the engineered tissue must initially challenge the inflammatory microenvironment of the damaged tissue to establish an efficient integration and, after the implantation, the inflammatory reaction characterizes a possible rejection process. The traditional strategy adopted after tissue and organ transplantation aims at minimizing the host immune response through anti-inflammatory and immunosuppressive therapies. However, if, on one hand, the immune system is often the cause of implants rejection, on the other, several immune components positively affect tissue regeneration and healing [4]. Therefore, controlling the balance between immune pro-inflammatory and pro-resolution players represents a better strategy to ensure implant tolerance than suppressing the immune response [5].

Among the immune cells involved in the foreign body response, macrophages play a central role. Macrophages are effector immune cells simplistically divided into two classes, named M1 and M2. To ensure regeneration, a balance between M1 and M2 activities (pro- and anti-inflammatory, respectively) shifting over time is required [6]; hence, scaffold-based approaches to direct macrophage polarization are gaining much interest in TE for regenerative medicine.

Nanomaterials are emerging as effective agents able to target macrophages, perturbing their polarization and thus their activity [7]. As a consequence, in recent years, optimally designed-nanoparticles (NPs) for the modulation of macrophage plasticity have been studied for treating diseases characterized by hyper-tolerant or inflammatory immune microenvironment, such as cancer [8] and inflammatory diseases [9], respectively. As far as TE is concerned, the full potential of nanoparticles as macrophage regulators has not yet been fully realized. This review aims at discussing nanotechnology impact on TE in terms of directing macrophage polarization and reprogramming, focusing on bioactive inorganic nanoparticles prospects for skeletal muscle regeneration.

## 2. The Immune Response to Implanted Materials: The Driving Role of Macrophages

The immune response against implanted materials originates as an acute inflammatory response to the tissue injury and the foreign material itself introduced with the implant [4]. Innate immunity is the first to be involved, providing an initial barrier against potential pathogens invading the damaged tissue. In the absence of pathogens, the innate response triggers a so-called sterile inflammation, put in motion by the danger signals released from the injured tissue, strongly affecting the outcome of regeneration. The innate response is then followed by the activation of the adaptive immunity. Although it was originally considered less important in the regeneration process, the adaptive immune response to implanted materials plays a pivotal role in tissue regeneration, mainly due to T cells activity [10]. In the following paragraphs, we will address the role of the innate and adaptive immune responses, focusing on macrophage intervention. The main steps are graphically represented in Figure 1.

### 2.1. Macrophages Orchestrate Innate Immunity

Tissue injury activates a local inflammatory response by the damage-associated molecular patterns (DAMPs) released from necrotic or stressed cells and damaged extracellular matrix (ECM). Danger signals typically include monosodium urate, extracellular ATP, nucleic acids, heat shock proteins (HSP), and fragments from ECM components, e.g., portions of collagen, hyaluronic acid, elastin, fibronectin, laminin, etc. Released DAMPs are then mainly recognized by Toll-like receptors (TLRs), initiating the inflammatory process via activation of the nuclear transcription factor NF-kB or interferon-regulatory factors. Upon TLR activation, tissue-resident macrophages are awakened to release chemoattractants for recruiting neutrophils, monocytes, and macrophages to the injury site, and to produce pro-inflammatory cytokines such as tumor necrosis factor-α (TNF-α), interleukin-6 (IL-6), and IL-1β [10].

Once crossed the endothelium and recruited to the implantation site, neutrophils produce antimicrobial substances and proteases to kill and degrade pathogens that have potentially invaded the damaged tissue. Furthermore, they also produce cytokines (e.g., IL-17) to recruit and activate more neutrophils, monocytes and macrophages, and growth factors (e.g., vascular endothelial growth factor, VEGF-A) to promote angiogenesis and proliferation of fibroblasts, epithelial cells, etc. [11]. Although being inflammatory cells, neutrophils also exhibit anti-inflammatory activity; indeed, recruiting macrophages, they promote their own removal, preserving the ECM from excessive degradation [6].

The recruited macrophages are differentiated from circulating monocytes and they often greatly exceed the population of tissue-resident macrophages involved in the very early immune response. Macrophages are key actors of tissue regeneration; indeed, they are responsible for the production of several cytokines, proteases, growth factors, soluble mediators, and ECM components, regulating tissue microenvironment [12]. Macrophages modulate the microenvironment and the latter in turn regulates macrophages activity; indeed, they may undergo several phenotypic and functional changes in response to different environmental stimuli [12]. Polarized macrophages can be simplistically classified in two main groups: classically activated macrophages (M1) activated by IFN-γ and lipopolysaccharides (LPS) and alternatively activated macrophages (M2), further divided in M2a (if activated by IL-4 or IL-13), M2b (by IL-1β or LPS) and M2c (by IL-10, TGF-β or glucocorticoids) [13]. This macrophage categorization based on specific activating factors *in vitro*, is, however, very simplistic, ignoring the source and the plurality of the stimuli coexisting in the in vivo environment [14]. In this regard, recent studies employing transcriptomic and proteomic analyzes have revealed a broader spectrum of interrelated macrophage activation states resulting from different sets of stimuli [15]. Coherently, macrophage polarization is presently the subject of debate. Nevertheless, here, for the sake of clarity, we use a bipolar nomenclature (M1 pro-inflammatory vs. M2 anti-inflammatory) to describe the two opposite sides of the spectrum of macrophage activation states. M1 macrophages exhibit pro-inflammatory and microbicidal activities, whereas M2 macrophages are anti-inflammatory immune cells, generally considered pro-resolution players, since they produce several ECM components and growth factors (e.g., VEGF-A) [12]. The kinetic of M1 and M2 macrophages is therefore crucial for tissue regeneration. In skeletal muscle regeneration, M1-related cytokines actively participate in skeletal muscle stem cells (satellite cells) activation. Afterwards, the M2-secreted cytokines IL-4 and IL-10 promote satellite cells proliferation and differentiation, increasing myogenin expression [16]. The mechanisms driving macrophage polarization towards the pro-resolution phenotype are still unclear; besides cytokines, microRNAs (miRs) are surely key regulators of macrophage polarization, via controlling the translation and degradation of messenger RNAs of cytokines and transcription factors [17,18]. MiR-146a, miR146b, miR222-3p, and miR181a, for example, are of particular interest for tissue regeneration, supporting M2 polarization in humans [19]. However, miRs-mediated macrophage regulation is out of the focus of this paper and a further study is under way from the same group.

### 2.2. Macrophage Role in Adaptive Immunity

Macrophages are not only part of innate immunity: they also prepare adaptive immunity to respond, generating costimulatory molecules for lymphocytes T and B activation by non-self-antigens, such as those occurring in materials carrying allogenic-cells or scaffolds not properly decellularized [4].

M2 macrophages trigger Th2-mediated response, generally having beneficial effects on tissue regeneration, whereas M1 macrophages promote strong T helper 1 (Th1) response, triggering pro-inflammatory cytokines production and cytotoxic (CD8^+^) T cells activation, impairing tissue healing. After bone injury, for example, Th1 and CD8^+^ T cells activation strongly correlates with impaired osteogenesis [20]. On the contrary, Th2 and regulatory T cells (Tregs) function as critical pro-resolution mediators; indeed, producing anti-inflammatory cytokines such as IL-10 and TGF-β, they promote at least bone [21], cardiac [22], and skeletal muscle [23] regeneration.

If on one hand it is evident that too sustained M1 macrophage activation can exacerbate inflammation increasing tissue injury, on the other, persistent M2 activation may also be counterintuitively detrimental, leading to pathological fibrosis [24]. Immunomodulation is therefore essential to trigger tissue regeneration, maintaining a delicate balance between the pro-inflammatory and the anti-inflammatory response.

## 3. Immunomodulation by Implanted Scaffolds

As reported in our previous work [4], materials properties are important for tissue regeneration to: (i) provide mechanical support to the implant, (ii) regulate scaffold adsorption/degradation time, (iii) promote stem cell recruitment and differentiation, and (iv) elicit a balanced immune response. In the following two paragraphs, we briefly examined the effects of native vs. synthetic scaffolds on macrophage polarization, discussing their impact on tissue regeneration.

### 3.1. Native Scaffolds: Decellularized ECM

Native scaffolds are highly bio-mimetic platforms, providing an ideal microenvironment for cell attachment, proliferation, and stem cell differentiation, having thus demonstrated to be the most appealing tool for TE applications [4]. They are obtained from several tissues and organs of different species by decellularization protocols. The resulting material is free of cells and cellular debris, but enriched with ECM components, such as collagen, fibrin, proteoglycans, and cryptic peptides, conferring it the positional and functional information required to re-build tissues and organs [5,25].

The immunomodulatory effect of decellularized ECM (dECM) mainly depends on scaffold structure and composition. Huleihel and colleagues, for example, recently demonstrated that matrix-bound vesicles (MBVs) present in urinary bladder matrix are responsible for macrophage polarization towards the anti-inflammatory phenotype, via three miRs (miR125b-5p, miR143-3p, and miR145-5p) carried within the vesicles [26]. The authors hypothesized that, during degradation of the ECM-based scaffold, the released MBVs are internalized by macrophages, eliciting their polarization.

The source tissue where the ECM is harvested is another factor determining the immune response to the scaffold. In this context, a recent study comparing macrophage activation after exposure to dECM derived from different tissues, highlighted heterogeneity in macrophage response due to distinct ECM degradation products. In particular, the exposure to urinary bladder matrix or small intestinal submucosa ECM differently affected murine bone marrow-derived macrophages, generating distinct gene expression profiles of several commonly investigated macrophage surface and activation markers [27]. Scaffolds deriving from the same tissue may also behave differently; dECM from skeletal muscle, for example, has been shown to trigger macrophage polarization towards the M2 anti-inflammatory phenotype in an in vivo model of skeletal tissue injury [28], but other studies reported no impact of skeletal muscle-derived ECM on macrophage plasticity [10]. Such a heterogeneity, even among matrices from the same tissue, may derive from the applied decellularization protocol. Indeed, even the detergent choice and treatment with specific enzymes to cleave immunogenic antigens may profoundly modify the immune response [29]. A recent work by Wu [30] showed that the application of a digestion-neutralization protocol to transform bone-derived filler particles into gel bio-scaffolds enhanced tissue regeneration in rat periodontal defect model [30]. Particularly, the applied approach shifted macrophage polarization towards the M2 phenotype, instead of the M1 pro-inflammatory phenotype generally triggered by the commonly used bone-derived ECM particles, enhancing the expression of pro-resolution mediators (e.g., IL-10) and decreasing the pro-inflammatory ones (TNF-α, IL-1β).

### 3.2. Synthetic Scaffolds

Synthetic scaffolds can be assembled from both naturally and synthetically-derived biocompatible polymers, in order to mimic the architecture and composition of several tissues and organs. Their molecular and structural characteristics can be finely controlled, with greater precision respect to their natural counterparts (i.e., dECM) [25,31]. Each material has different physico-chemical and biological properties, often contributing to a great extent in eliciting the foreign body response, which therefore becomes a critical issue for successful scaffold implantation [2].

As previously mentioned, different materials trigger diverse immune responses. In a recent work by Badylak’s group, the hydrogelating self-assembling fiber (hSAF) system, formed from two complementary de novo designed α-helical peptides, strongly modulated macrophage plasticity, increasing the expression of anti-inflammatory markers such as Arginase1, IL-10, and CD206, finally contributing to remodeling and myogenic differentiation [32]. Chitosan also polarized human macrophages towards the M2 phenotype, downregulating the expression of pro-inflammatory TNF-α, while increasing the levels of the anti-inflammatory IL-10 and TGF-β [33], whereas poly-ethylen-glycol (PEG)-based hydrogels enhanced pro-inflammatory cytokines expression in murine RAW 264.7 macrophages [34].

Material-induced immune response is even more complicated, as the same material can divergently stimulate specific immune cell populations. This is the strange case of chitosan, that, on the one hand, triggered macrophage polarization towards the anti-inflammatory subtype, and, on the other, polarized dendritic cells (DCs) towards a pro-inflammatory type [33].

Besides composition, other crucial factors regulating the immune response concern the structural properties of the material (e.g., porosity, pore distribution), surface topography, stiffness, etc. [25]. Regarding the impact mediated by material porosity, Tylek and colleagues recently exploited advanced melt electrowriting (MEW) to fabricate poly-ε-caprolactone (PCL)-based scaffold with box-shaped pores with various inter-fiber spacing, from 100 μm to 40 μm. The resulted scaffolds facilitated human primary macrophages elongation and polarization towards the M2 phenotype, with the smallest pore sized as the most effective [35]. Scaffold surface topography is widely recognized as modulator of the immune players; in the last decade, indeed, many studies highlighted the possibility to attenuate the inflammatory process and drive macropages towards the pro-healing phenotype [36], by controlling scaffold surface grooves and gratings—for example, via 3D lithographic methods. These approaches allow for finely regulating a scaffold micro- and nano-pattern, thus strongly affecting macrophages elongation and differentiation [37].

Scaffold stiffness, instead, is generally considered less important in driving macrophage polarization, especially if compared to scaffold architecture [25]; however, recent studies strongly suggest that substrate stiffness can make the difference. Sridharan and colleagues, for example, varying collagen-coated polyacrylamide gels stiffness, were able to alter human THP-1-derived macrophages behavior in terms of migration mode, polarization state, and, thus, a functional role. Particularly, the highest stiffness (323 kPa) primed macrophages towards the M1 phenotype, whereas the softest (11 kPa) towards the M2 phenotype [38]. The same authors further demonstrated that the variability in macrophage polarization in response to differential stiffnesses may be dependent on the crosslinking agents used to modulate collagen stiffness [39].

It is therefore evident that the macrophage response induced by scaffolds (both natural and synthetic) is highly variable, mainly because there are many factors involved (biomaterial, composition, topography, porosity, etc.). Furthermore, it is arduous to recreate the complexity of the microenvironmental stimuli naturally polarizing macrophages; thus, novel approaches such as microfluidic systems are trying to recreate in vitro, the in vivo stimulation [40].

Altogether, there is a need to find new strategies more finely controlling macrophage plasticity, to help scaffolds addressing the immune response towards tissue regeneration. Nanotechnology may represent the solution.

## 4. Nanoparticles to Direct Macrophage Polarization

Nanoparticles (NPs) have been initially developed to overcome problems of bioavailability, body retention, solubility, stability, and selectivity of pharmaceutical agents, protecting the carried drug until reaching the desired body district. However, nanomaterials at the nanoscale (1–100 nm) acquired peculiar properties, due to the increased reactive surface/bulk ratio with respect to micro- and macro-structures [41], making them attractive for TE.

Indeed, in the 2000s, a key role has been recognized for nanomaterials in TE, as nanocomposite polymers, both in the form of electrospun fibers and hydrogels, often provide superior mechanical, functional, and electrical properties [42]. In the context of skeletal muscle regeneration, for example, aligned nanofibrous scaffolds (e.g., PCL/collagen) favored cell alignment and myotube formation, thus promoting muscle regeneration [43].

It is noteworthy that the role of NPs as modulators of macrophage plasticity is strongly emerging; indeed, due to their particulate (instead of molecular) nature, nanoparticles preferentially target professional phagocytes, such as macrophages [44]. This property is of a paramount importance for the success of TE procedures. Therefore, several research works focused on the use of organic nanovesicles (liposomes, polysaccharides, capsules, etc.) carrying encapsulated bioactive molecules [45], such as flavonoids [46], miRs, and cytokines [24], in order to regulate macrophage activity. In this study, instead, we investigate three major inorganic NPs (gold, titanium oxide, and cerium oxide NPs, later denoted as AuNPs, TiO_2_ NPs, and CeO_2_ NPs), not only being exploitable as carriers for drugs and molecules, but also being characterized by intrinsic bioactivity, which renders them promising candidates for TE therapies (Figure 2, Table 1).

### 4.1. Harnessing Gold Nanoparticles to Trigger Skeletal Muscle Regeneration

AuNPs have been extensively studied [55], particularly for cardiac tissue regeneration, due to their ability to enhance scaffold conductivity [42]. Furthermore, they were recently found to improve osteogenic differentiation of human bone marrow-derived mesenchymal stem cells [56]. As regards their immunomodulatory activity, a recent work by Ni highlighted the ability of AuNPs to generate a microenvironment with constraint inflammatory and reparative cytokines [57]. In particular, when facing inflammation upon stimulation with LPS, murine macrophages exposed to AuNPs showed decreased pro-inflammatory and enhanced anti-inflammatory markers, with respect to control cells.

The surface chemistry of AuNPs also allows rapid functionalization with bioactive molecules, making them suitable for several purposes. In the context of macrophage polarization, AuNPs functionalized with the peptide arginine-glycine-aspartic acid (RGD), a motif responsible for cell adhesion to ECM, were observed to dramatically decrease M1 markers expression, while increasing the M2 ones, in macrophages isolated from mouse livers [47]. Furthermore, when coated with the hexapeptides Cys-Leu-Pro-Phe-Phe-Asp, able to inhibit LPS-induced activation of TLR4, AuNPs drove mouse bone marrow-derived macrophages towards the M2 phenotype, both in vitro and in vivo [48]. Cytokines may also be conjugated on nanoparticles surface; a recent work by Raimondo and Mooney, reported that IL-4-conjugated AuNPs recovered injured murine skeletal muscle via NP-directed M2 macrophage polarization [49]; indeed, the authors finely demonstrated that macrophage depletion abrogated the therapeutic effect triggered by the functionalized nanoparticles. The potential positive contribution of AuNPs in skeletal muscle regeneration is confirmed by Ge and colleagues. Indeed, they reported that AuNPs significantly enhanced myogenic differentiation of myoblasts, via activation of the p38 mitogen-activated protein kinase (p38 MAPK) signaling pathway, finally promoting in vivo regeneration in muscle defect models of rats [58].

### 4.2. Titanium Oxide NPs: From Osteogenesis to Muscle Regeneration?

TiO_2_ NPs are of significant concern in TE, especially for hard tissue regeneration, because titanium resists corrosion from the body. However, nanometer-thin TiO_2_ deposition has also been applied for muscle regeneration, significantly enhancing rat skeletal muscle cell attachment and proliferation [59]. Unlike other ceramic NPs, which induced an inflammatory phenotype in primary macrophages, titanium oxide NPs were reported not to foster inflammation [60]. A recent work demonstrated that TiO_2_ nanotubes promoted osteogenesis via mediating the crosstalk between macrophages and mesenchymal stem cells (MSCs) under oxidative stress, reducing early inflammation by accelerating macrophages M1 to M2 transition [50]. A previous work by Lee, also reported increased M2 subtypes markers (Arg1, MR and CD163) on calcium and strontium (Ca and Sr, respectively)-modified titanium implants, leading to enhanced osteogenesis [51]. Furthermore, TiO_2_ nanotubes, inducing macrophage elongation and polarization towards the M2 anti-inflammatory state, also promoted endothelialization [52]. In particular, THP-1 cells, when exposed to TiO_2_ nanotubes, polarized to M2 macrophages and produced high levels of VEGF, which promoted human umbilical vein endothelial cells (HUVEC) proliferation via two main proliferative pathways: the phosphatidylinositol 3-kinase/AKT and the extracellular signal-regulated kinase 1/2 pathways. As vasculature is needed to facilitate construct engraftment and provide nutrient and waste exchange to the regenerating tissue, TiO_2_ nanoparticles may be able to strongly affect the whole foreign body response, addressing it towards regeneration.

### 4.3. Cerium Oxide NPs: Not Just Antioxidants

CeO_2_ NPs are bioactive nanoparticles attracting great interest in the biomedical field, mainly for their strong antioxidant activity, due to the presence on their surface of cerium atoms in both of the oxidation states: Ce^3+^ and Ce^4+^ [41]. Besides their valence in cancer research [61], they have recently been emerging as promoters of tissue regeneration. In 2013, topical application of CeO_2_ NPs was reported to accelerate the healing of dermal wounds in mice, enhancing proliferation and migration of fibroblasts, keratinocytes, and vascular endothelial cells [62]. More recently, CeO_2_ NPs incorporated into hydroxyapatite coating, via the plasma spraying technique, induced polarization of murine RAW264.7 macrophages towards the M2 phenotype, as demonstrated by the upregulation of the M2 surface markers CD163 and CD206, promoting an osteogenic behavior of bone mesenchymal stem cells (BMSCs) [53]. The improved osteogenic differentiation ability of BMSCs was further found to be proportional to the surface Ce^4+^/Ce^3+^ ratio; indeed, the higher ratio achieved better osseointegration in vivo, and enhanced murine macrophages polarization towards the M2 phenotype, increasing the levels of anti-inflammatory cytokines [54].

It is noteworthy that CeO_2_ NPs demonstrated reliable biocompatibility [63]. This is of a paramount importance; indeed, in general, pharmacological administration of inorganic nanoparticles causes safety issues to arise, rendering nanotoxicology [64] a key discipline for nanoparticles use in biomedicine and TE.

## 5. Conclusions

Despite the significant progress in TE applications, nanotechnology (both in terms of nanocomposite polymers and/or nanopatterned scaffolds) has been mainly applied to provide scaffolds with superior mechanical, functional, and electrical properties, and to favor muscle cell alignment and differentiation. Nevertheless, as thoroughly discussed in this review, particular attention should be paid to the induced immune response, especially to macrophage polarization. Indeed, a correct balance between pro- and anti-inflammatory factors is crucial for scaffold engraftment and tissue regeneration. In this context, surprisingly, there is still too little effort to investigate the contribution that nanotechnology may give to direct macrophage polarization, while, in other areas of biomedical research, such as inflammatory diseases and tumors, several bioactive inorganic nanoparticles have been recognized as valuable modulators of macrophage plasticity.

The analysis of the literature performed in this review unveiled the unpresented potential of Au, TiO_2_, and CeO_2_ nanoparticles to effectively modulate macrophage polarization and strongly support TE for regenerative medicine. While AuNPs have already been studied in the field of muscle regeneration, obtaining excellent results both in terms of NP-elicited macrophage polarization and promotion of myogenesis, TiO_2_, and CeO_2_ NPs, were mainly tested in the field of osteogenesis, but the results summarized here represent good premises for their immediate application also in skeletal muscle regeneration.

## Figures and Tables

**Figure 1 nanomaterials-10-01963-f001:**
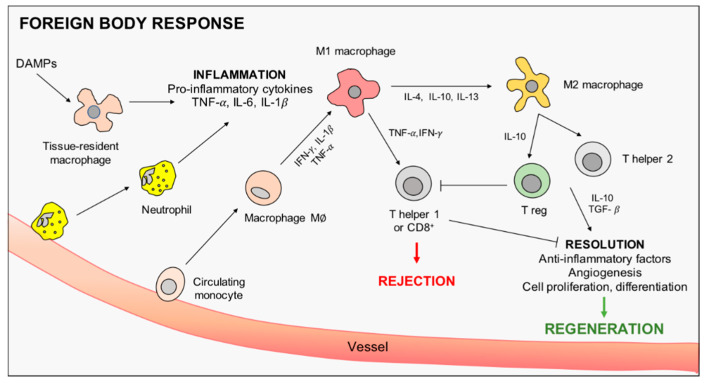
Graphical representation of macrophage intervention in scaffold-induced foreign body response. From the left, danger associated molecular patterns (DAMPs) are released after tissue injury, activating tissue-resident macrophages to promote inflammation and neutrophils recruitment. Inflammation is then sustained by pro-inflammatory M1 macrophages, differentiated by circulating monocytes, before being eventually resolved by macrophage polarization towards the M2 anti-inflammatory phenotype. Macrophage-driven stimulation of T cells is also shown.

**Figure 2 nanomaterials-10-01963-f002:**
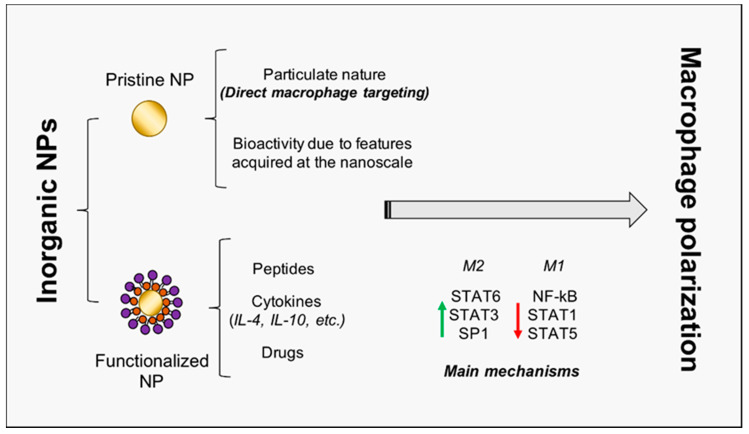
Inorganic NP-mediated macrophage polarization towards regeneration. The schematic representation of inorganic NPs double role is reported. NPs may act as pristine NPs, characterized by intrinsic bioactivity, and as carriers for peptides, cytokines, and drugs. Principal transcription factors involved in macrophage activation and polarization are also included. NP = nanoparticle; IL = interleukin; STAT = signal transducer and activator of transcription; NF-kB = nuclear factor kappa-light-chain-enhancer of activated B cells.

**Table 1 nanomaterials-10-01963-t001:** Nanoparticle-induced macrophage polarization towards M2 phenotype.

NPs	Shape	Size (nm)	Surface Modification	Initial Phenotype	Polarized Phenotype	Model	Used Markers	Clinical Aim	Ref.
**Au**	rods	ND	PEGylation + RGD	M∅	M2	Mouse	Arg-1, IL-4, TNF-α, Retnla	Acute hepatitis	[47]
	spheres	13	hexapeptides	M2	M2	Mouse BMDMs	IL-12, IL-6, IL-10, iNOS, Arg-1, YM1	Acute lung injury	[48]
				M1	M2				
					M2	ALI Mouse model	CD80, CD206		
	spheres	30	PEGylation + IL-4	M∅	M2	Human THP-1 cell line	CD206, CD163, IL-4	Muscle recovery	[49]
		30	PEGylation + IL-4	M∅	M2	Mouse	CD206, CD80		
		100	None	M∅	M1	Human THP-1 cell line	CD206, CD163, IL-4		
**TiO_2_**	tubes	110	None	M∅	M1	Mouse RAW 264.7 cell line	CCR7, IL-6, IL-10, VEGF, BMP2, TGF-β1	Osteogenesis	[50]
				M∅	M2	Mouse RAW 264.7 cell line + MSCs			
	disks	10 *	Ca^2+^, Sr^2+^	M∅	M2	Mouse RAW 264.7 cell line	Arg-1, CD163, CD86, iNOS	Osteogenesis	[51]
	tubes	65 92 142	None	M∅	M2	Human THP-1 cell line	Arg-1, CD206, IL-10, VEGF	Heart valve replacement	[52]
**CeO_2_**	thin layer	ND	None	M∅	M2	Mouse RAW 264.7 cell line	CD163, CD206, IL-6, TNF-α, IL-10, TGF-β1	Osteogenesis	[53]
	thin layer	ND	None	M∅	M2	Mouse RAW 264.7 cell line + BMSCs	CCR7, CD206, TNF-α, IL-10	Osteogenesis	[54]
		ND	None			Rat			

NPs = nanoparticles; RGD = arginine-glycine-aspartic acid; M = macrophage; Arg-1 = arginase-1; IL = interleukin; TNF-α = tumor necrosis factor-α; MSCs = mesenchymal stem cells; BMDM = bone marrow derived macrophages; CCR7 = C-C chemokine receptor 7; VEGF = vascular endothelial growth factor; BMP2 = bone morphogenetic protein 2; TGF-β1 = transforming growth factor β1; NOS = nitric oxide synthase; BMSCs = bone mesenchymal stem cells; * Average surface roughness; ND = not determined.

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
