# Peer review of "Harnessing Inorganic Nanoparticles to Direct Macrophage Polarization for Skeletal Muscle Regeneration"

_nanomaterials, 2020, doi:10.3390/nano10101963_

Round 1

Reviewer 1 Report

  • The review is about nanoparticle for Macrophage polarization. The authors, through whole review, discuss 2 classes of macrophages M1 and M2, and possibility of inorganic NPs to direct macrophage polarization. Nevertheless, I think up to date review should at least mention that M1, M2 polarization model of macrophages are in question at the moment. Following publications (https://doi.org/10.1016/j.immuni.2014.01.006 and doi: 10.12703/P6-13 and doi:10.1038/nn.4338 ) revealed a spectrum of macrophage activation states extending the current M1 versus M2-polarization model. Current review doesn’t say anything about that.
  • please in the chapter - Synthetic scaffolds - mention also 3D lithography structures and microfluidic systems. There were several works on that.
  • Please add more references on the topic : Nanoparticles to direct macrophage polarization

Author Response

Dear reviewer,

we appreciated your comments and suggestions, and amended the text according to them (yellow part).

Here below our answers, point by point:

  1. Nevertheless, I think up to date review should at least mention that M1, M2 polarization model of macrophages are in question at the moment. Following publications (https://doi.org/10.1016/j.immuni.2014.01.006 and doi: 10.12703/P6-13 and doi:10.1038/nn.4338) revealed a spectrum of macrophage activation states extending the current M1 versus M2-polarization model. Current review doesn’t say anything about that.

Your suggestion is right; indeed, we added sentences on the complexity of macrophage activation states, highlighting that the M1/M2 polarization model is too simplistic (lines 107-113*, page 3).

  1. Please in the chapter - Synthetic scaffolds - mention also 3D lithography structures and microfluidic systems. There were several works on that.

In the amended text we reported additional information on 3D lithography and microfluidics within the “Synthetic scaffolds chapter” (lines 210-215 page 5, and 227-229, page 6).

  1. Please add more references on the topic: Nanoparticles to direct macrophage polarization

More references on the macro-topic “Nanoparticles to direct macrophage polarization” have been discussed, increasing the balance between the two macro-topics addressed (macrophage and nanoparticles).

*Please note that lines may be different according to the operating system and the opening of comments

Reviewer 2 Report

Dear authors,

 This review is very well written, and clearly explained the mechanism of macrophages polarization which influenced by bioactive inorganic nanoparticles for skeletal muscle regeneration.

 I think that this review will be clearer if authors explain the mechanism with a little more graphical representaion.

Author Response

“This review is very well written, and clearly explained the mechanism of macrophages polarization which influenced by bioactive inorganic nanoparticles for skeletal muscle regeneration. I think that this review will be clearer if authors explain the mechanism with a little more graphical representation.”

Dear reviewer,

Thank you for your favorable comments.

We greatly appreciated your suggestion and indeed, added a new figure (Figure 2, page 7) to better elucidate the mechanisms of macrophage polarization by inorganic nanoparticles.

Reviewer 3 Report

dear authors, this is a well-written review on first, the immune response to implants with a focus on M1/M2 macrophages, second the role of scaffolds in steering this M1/M2 ratio and third the effects of nanoparticles on this ratio. While the manuscript is well-written the content which the Title alludes to is, unfortunately, only a single page (paragraph 4.1) out of the six pages of text. I missed a discussion on nanopatterned implants as this is clearly much more promising than metal nanoparticles. Finally, I missed an overall take-home message.

Some suggested improvements (minor comments):

lines 157-160: it could be added how dECM from different tissues affects the macrophage response

line 160: "scaffolds may behave differently": two studies are compared, there may be differences in methods between the two studies

line 197: 100 uM to 40 uM: 100 um to 40 um?

Table 1: why not indicate the size(s) of other shapes than spheres: rods, tubes, disks, thin layer?

;should read um

techniques b

Author Response

Dear reviewer,

We greatly appreciated your comments and suggestions, and amended the text according to them (green part), being sure they have given an extra value to the proposed review.

Here below our answers, point by point:

  1. While the manuscript is well-written the content which the Title alludes to is, unfortunately, only a single page (paragraph 4.1) out of the six pages of text. I missed a discussion on nanopatterned implants as this is clearly much more promising than metal nanoparticles.

Your concern is right; in this regard, we extended the macro-topic “Nanoparticles” (pages 6-10), discussing more references as much as we could: the novelty and particularity of the addressed topic and the proposed approach, limit the extension of the paragraph, as few published works deal with the topic.

The nanometric modulation of scaffold patterns is certainly a very promising strategy to stimulate the differentiation of muscle tissue, and partly for the modulation of the immune response, and thus we included hints on the potentiality of nanopatterned implants (lines 210-215*, page 5, yellow paragraph). As far as the title is concerned, we modified it according to your comment; the amended one, better reflects the focus of the proposed review, that is to highlight the potential use of inorganic nanoparticles (metallic and ceramics) in the modulation of macrophage activity.

  1. I missed an overall take-home message.

In order to better elucidate the take-home message, we have modified the conclusions (pages 10) and slightly the abstract (page 1). We think the lesson that can be learned from our review is now clearer (emended manuscript).  

Some suggested improvements (minor comments):

  • lines 157-160: it could be added how dECM from different tissues affects the macrophage response

We added some insights on this subject (lines 167-171, page 4).

  • line 160: "scaffolds may behave differently": two studies are compared, there may be differences in methods between the two studies

There are indeed differences in both the methodologies and models used; further details are given later in the text.

  • line 215: 100 uM to 40 uM: 100 um to 40 um?

Right, we fixed it. Thanks

  • Table 1: why not indicate the size(s) of other shapes than spheres: rods, tubes, disks, thin layer?

Sizes not reported in Table 1 are due to not determined sizes in the corresponding paper; thus, we added the note ND in the Table 1.

*Please note that lines may be different according to the operating system and the opening of comments

Round 2

Reviewer 1 Report

Only Spellcheck required.

Author Response

Dear reviewer,

we performed a spell check as requested.